



# The monsoon hydroclimates in HadGEM3 model configurations GA3.0 and GA4.0: Impact of remote versus local circulations errors and horizontal resolution

Wilfran Moufouma-Okia[1,4], Debertini A. Vondou[2], Richard Jones[3]

[1]Regional Climate Predictions services Division, World Meteorological Organization (WMO), Geneva, Switzerland
[2]Laboratory for Environmental Modelling and Atmospheric Physics, University of Yaoundé 1, Yaoundé, Cameroon
[3]Met Office Hadley Centre, Exeter, United Kingdom
[4]Université Paris-Saclay, Technical support unit of IPCC Working Group 1, Saint Aubin, France

*Correspondence to*: Wilfran Moufouma-Okia (wilfranmo@gmail.com)

**Abstract.** State-of-the-science general circulation models (GCMs) are the primary tools for making policy-relevant climate calculations. Yet, these models face challenges in monsoon regions where live more than 70% of the world's population, due to the complex interplay of local and remote influences on a spectrum of space and time scales. This work examines the fidelity to reproduce regional and global monsoons climatological features using the Met Office Unified Model (MetUM) third and fourth generations Global Atmosphere (GA3.0) and (GA4.0), two configurations of the HadGEM3 system developed for seamless use across climate and weather time scales. Results are compared both against multiple observational gridded datasets and outputs from 20 atmospheric-only GCMs simulations from the CMIP5 campaign. Furthermore, we investigate the influence of remote versus local atmospheric circulation errors by constraining realistically HadGEM3 circulations over a prescribed monsoon domain and examining the consequences outside and inside this domain using the "grid-point nudging" method. The GA3.0 largely captures global monsoon features, including the monsoon precipitation patterns and extent of regional monsoon domains, when integrated using a low (~135-km in mid-latitudes), medium (~60 km in mid-latitudes) and high (~25km in mid-latitudes) horizontal resolutions. GA4.0 and GA3.0 results display a close similarity, and compares reasonably well against the best available CMIP5 models. The common failure of HadGEM3 configurations is the simulated weak magnitude and extent of the Asian Summer Monsoon (ASM) precipitation pattern, and associated low-level Somali jet. This situation is also apparent within HadGEM2-A, ACCESS1-0, and CSIRO-Mk3-6-0 – three GCMs sharing dynamical and physical components. HadGEM3 performance improves significantly over ASM with atmospheric circulations constrained realistically over the tropics, West African and Asian Summer monsoon domains. Conversely, constraining atmospheric circulations over other remote monsoon domains show little influence on the ASM precipitation. We argue that GA4.0 and GA3.0 poor simulations over the ASM domain are attributable, partly to local atmospheric circulations errors and excessive





precipitation over the southwest equatorial Indian Ocean, rather than to remote tropical atmospheric responses of varying forcing fields, such as SST over the Arabian Sea, aerosols, and growing greenhouse gas emissions. The improved understanding of GCM performance in monsoon regions is an important step for credible projections of global monsoon changes.

KEY WORDS: Global monsoon, atmospheric only general circulation model, model systematic errors, MetUM, model evaluation.

**1 Introduction**

The prediction of monsoonal changes in the context of global warming is a critically important societal challenge for
sustainable development, planning and  disaster risk reduction, particularly for 20% of the Earth's surface, where live more than 70% of the world's population (Sooraj et al., 2015; Zhisheng et al., 2015). The Intergovernmental Panel on Climate Change (IPCC) Fifth Assessment Report (AR5) concludes that under high-emission scenarios, the collective effect of monsoon systems worldwide is likely to strengthen in the future, with increase in its area coverage and intensity, while there is a decreasing trend of lower-troposphere monsoon circulations (Christensen et al., 2013; Hsu et al., 2013; Kitoh et al., 2013). The
increase of monsoon coverage will be stronger in the Northern Hemisphere (NH) with lengthening of rainy seasons,  compared to the Southern Hemisphere (SH), due to earlier onset dates and a delayed retreat (Lee and Wang, 2014; Ha et al., 2020a).  The increase in moisture flux convergence and surface evaporation are considered the most probable causes behind  strengthening of the monsoon rainfall, resulting in the "wet-get-wetter" pattern (Richter and Xie, 2008; He and Soden, 2017; Deng et al., 2018). However, projected monsoonal changes indicate a region-dependent and uncertain responses under the 1.5°C or 2°C
global warming above the pre-industrial levels, as proposed by the 2015 Paris Agreement, though robust differences in temperature means and extremes are expected (Tschakert, 2015; United Nations Framework Convention on Climate Change, 2015; Masson-Delmotte et al., 2018b, 2018a; Qu and Huang, 2019).

The monsoons are central components of the climate system and important conveyors of the atmospheric moisture and energy
at the global scale, occurring notably in the tropics and exhibiting significant seasonal reversal of lower-tropospheric circulations with clear alternation between wet summers and dry winters (Zhisheng et al., 2015). There are several viewpoints explaining the monsoons dynamics and thermodynamics including the equilibrium between net radiative energy and the transport of moist static energy, associated with the seasonal changes in the large-scale contrast between heat capacities over land and upper ocean (Webster et al., 1998; Chang et al., 2009; Liu et al., 2009; Wang, 2009; Wu et al., 2012; Wang et al.,
2013, 2014; Zhisheng et al., 2015; Mohtadi et al., 2016; Ha et al., 2020b). Monsoon changes are affected by interactions with the slowly varying boundary conditions such as sea surface temperature (SST), large-scale atmospheric and ocean circulation



patterns, remote impact of large-scale atmospheric modes of variability, changes in the Walker circulation, seasonal migrations of the intertropical convergence zone, and regional physiographic details including orography and land cover (Grimm, 2004; Kullgren and Kim, 2006; Raia et al., 2008; Zhang et al., 2009; Marengo et al., 2012; Turner and Annamalai, 2012; Wang et al., 2013; Boone et al., 2016; Cerezo-Mota et al., 2016; Halder and Dirmeyer, 2016; Li et al., 2016; Wüthrich et al., 2016; He et al., 2017; Pathak et al., 2017; Roy et al., 2017; Smith, 2018; Grimm, 2019; Monerie et al., 2019). Anthropogenic aerosols and land-atmosphere interactions related to land use and land cover changes (LULCC) can also have significant biophysical impacts on monsoon dynamics (Hosseinpour and Wilcox, 2014; Li et al., 2016; Lau and Kim, 2017; Quesada et al., 2017).

Recent scientific advances have extended the monsoon perspective from regional to global scales using the concept of Global Monsoon (GM) and efforts have been devoted to clarify the monsoon connections with tropical and extra-tropical climate processes (Kitoh et al., 2013; Dike et al., 2015; Chang et al., 2018; Deng et al., 2018). The GM illustrates the variability in the global-scale overturning atmospheric circulations, associated with the combined effect of regional monsoon systems worldwide, and how taken together, these systems dominate the global transport of atmospheric energy and water vapour (Trenberth et al., 2000; Kim et al., 2008; Chang et al., 2009; Hsu et al., 2011; Wang et al., 2011; Wang et al., 2012; Christensen et al., 2013; Lee et al., 2014; Lin et al., 2014; Zhisheng et al., 2015; Chang et al., 2018; Deng et al., 2018; Jiang and Zhou, 2019; Ha et al., 2020b). The features of GM include global homogeneity, regional diversity, seasonality, instability, and a domain that is delineated by the annual range (local summer-minus-winter) of precipitation that exceeds 2.5 mm/day and 55-70% of the local annual mean rainfall (Lin et al., 2014; Wang et al., 2014). According to this definition, dominant monsoon systems span over continents and oceans and include the South American monsoon (Marengo et al., 2012), North American monsoon (Barlow et al., 1998), Indian or South Asian monsoon (Ramesh and Goswami, 2007), South China Sea monsoon (Luo et al., 2016), Indochina Peninsula (Qian and Jiang, 2016), Australian monsoon (Kullgren and Kim, 2006), South African monsoon, West African monsoon (Redelsperger et al., 2006), East Asian-western North Pacific monsoon (Ha et al., 2017), and Maritime Continent monsoon (Webster et al., 1998) subsystems. Multiple lines of evidence indicate a decreasing trend for the GM precipitation intensity and monsoon coverage over land in the period 1948-2003, with primary contributions from the weakened West African and South Asian monsoon systems associated with anthropogenic forcing and internal variability (Wang and Ding, 2006). GM precipitation in one hemisphere is also enhanced through a remote "volcanic forcing-induced-intensification" mechanism and atmospheric circulations changes (Liu et al., 2009; Min et al., 2013; Liu et al., 2016). Linkages between ENSO and precipitation emerge as an important feature of the Australian monsoon and the strong seasonal cycle of precipitation is influenced by the surrounding oceans (Evans et al., 2014; Gallego et al., 2017). Monsoon precipitation changes in North and South America have also been connected to SSTs (Vera et al., 2006; Marengo et al., 2012; de Carvalho and Cavalcanti, 2016; Pascale et al., 2016, 2017). In Africa, the severe and long-lasting Sahel drought, which plagued the region from the 1970s to the 1980s is the most distinctive monsoon feature (Biasutti and Giannini, 2006; James et al., 2013; Chadwick et al., 2017b; Vizy and Cook, 2017; Cook and Vizy, 2019). The West African monsoon climate usually interacts with monsoon



systems from remote regions, suggesting that both regional and global perspectives are needed to improve the understanding and predictions of monsoon climates (Bielli et al., 2010; Flaounas et al., 2012; Rowell et al., 2013; Chadwick et al., 2017a).

Climate simulations performed with Coupled General Circulation Models (CGCMs) are the primary tools for assessing the Earth system's response to large-scale forcings and developing policies to address the future global climate change through
hypothesis testing, sensitivity studies, internationally coordinated multi-model experiments and evaluation campaigns. The Coupled Model Intercomparison Project Phase 5 (CMIP5) and 6 (CMIP6) provide the basis for multimodel evaluation and reveals a variety of systematic differences between models and observations, with many persisting from a model generation to the next (Sperber et al., 2013; Eyring et al., 2016b; Annamalai et al., 2017; Anand et al., 2018; Xue et al., 2018). The evaluation of historical simulations and estimation of uncertainty in the future projections are prerequisite to assessment of the
future changes. Confronting CGCMs outputs against observations provides further insight into shortcomings and ways in which the various processes are represented within climate models. Several studies suggest that the CMIP5 multi-model ensemble mean captures well observed spatial patterns and interannual variability of GM's rainfall, though most individual models underestimate its extent and intensity (Flato et al., 2013; Wang et al., 2017). CMIP6  models simulate  better global monsoon  intensity  and precipitation over CMIP5 models,  but common biases  and large  intermodal  spreads persist (Wang
et al., 2020). Moreover, projected CMIP6 monsoon precipitation indicates that an important uncertainty arises from  circulation changes, which can be partly explained by model-dependent response to uniform sea surface temperature warming (Chen et al., 2020).

There are long standing errors in the CMIP5 sea surface temperature (SST) field leading to insufficient South Asian monsoon
rainfall along with too weak monsoon circulations (Annamalai et al., 2017; Găinuşă-Bogdan et al., 2018); systematic southward shift of the West African monsoon (WAM) rainfall (Roehrig et al., 2013); unrealistic simulations of easterly low-level moisture flux across the Caribbean region, which can lead to  the well-known "monsoon retreat bias" or excessive North American monsoonal rainfall in the fall (Pascale et al., 2016, 2017); and deficiencies in simulating the East Asian Summer monsoon rainfall including topography-related cold biases, excessive rainfall, unrealistic southeast–northwest rainfall gradient,
overestimated interannual variability of temperature and precipitation, and unrealistic monsoon circulation's magnitude (Boo et al., 2011; Feng et al., 2014; Li et al., 2014). The analysis of 28 CMIP5 models highlights issues to resolve the spatial and temporal characteristics of the South Asian Monsoon precipitation adequately and it is found that precipitation and 2 m surface air temperature (T2m)  biases are reduced with increase in the model horizontal resolution (Anand et al., 2018). Furthermore, model systematic errors are often reduced for atmosphere-only models (AGCMs) simulations forced by observed historical
sea SSTs, suggesting that many CGCMs biases are not inherent and arise from the challenge of capturing interactions between monsoon systems and oceanic processes (Wang et al., 2017; Jiang and Zhou, 2019).

This paper is motivated by the opportunity to advance our understanding on GCM simulations of the monsoon hydroclimates. The main purpose is to assess the capability of the Met Office Unified Model (MetUM) Global Atmosphere (GA) modelling

systems GA3.0 and GA4.0, two configurations of the HadGEM3 system employed for seamless predictions across climate and weather time scales. For this we attempt to disentangle the relative influence of physical formulation, horizontal resolution, and remote versus local atmospheric circulations on the HadGEM3system's performance, relative to that of state-of-the-science experiments from the CMIP5 dataset. This analysis provides a  framework for further understanding the interactions between individual monsoon systems and associated mechanism including teleconnections, though it may not address fully

the issue of attributing specific causes to specific systematic errors. We argue that this is crucial to improve the credibility of climate predictions and projections of future GM changes, and to increase the confidence that model development chains seen to improve performance in individual monsoon region are doing so by modelling accurately climate processes underpinning the GM variability. Thus, refining the understanding of climate model's uncertainty is an important step to mitigate the impact of monsoon-dominated climate extremes on human and natural systems, for a large population, mostly in developing countries

(Cerezo-Mota et al., 2016; Novello et al., 2017; Chang et al., 2018; Huang et al., 2018; Mishra et al., 2018; Talento et al., 2020).

The remainder of this article is organized as follow: Sect. 2 describes the data, models, method of analysis and experimental design. Sect. 3 assesses the performance of GA3.0 and GA4.0 configurations relative to CMIP5 experiments and analyses the

influence of remote versus local atmospheric circulation errors on the model performance, through application of the "grid-point nudging" or "Newton relaxation" strategy.  Concluding remarks are provided in Sect. 4.

## 2 Data and methods

### 2.1 Models

This study uses two configurations of the Met Office Unified Model HadGEM3 system, formulated by converging the

development paths of the weather and climate global atmospheric model components and whereby atmospheric processes are modelled seamlessly across spatial resolutions and time scales, notably the Global Atmosphere 3.0 (GA3.0) and 4.0 (GA4.0) components (Walters et al., 2011, 2014). The usage of "seamless modelling" refers to predicting the spatial–temporal continuum of the interactions among weather, climate, and the Earth system (Palmer et al., 2008; Hurrell et al., 2009; Ruti et al., 2020).

The GA3.0 configuration uses a semi-implicit semi-Lagrangian dynamical core to solve the non-hydrostatic, fully compressible deep atmosphere equations of motion (Davies et al., 2005). Dry atmospheric prognostics are discretized horizontally onto a regular longitude/latitude grid with Arakawa C-grid staggering  (Arakawa and Lamb, 1977) and include




the three-dimensional wind components, potential temperature, Exner pressure, and density. Moist prognostics such as specific humidity, prognostic cloud fields, and other atmospheric loadings are advected as free tracers. The vertical decomposition is

performed using terrain-following hybrid height coordinates and Charney-Phillips staggering approach (Charney and Phillips, 1953). Large scale clouds are considered using prognostic cloud fractions and prognostic condensate (Wilson et al., 2008) and hydrology-soil-vegetation- atmosphere interactions are calculated using the Global Land 3.0 (GL3.0) Joint UK Land Environment Simulator (Best et al., 2011).

The GA4.0 version uses dynamical and physical cores that are largely identical to GA3.0. But, it has a slightly advanced
formulation of sub-grid scale processes including the replacement of specific quantities for moist prognostics with mass mixing ratios; a correction to the treatment of shortwave fluxes in the coupling to the sea ice component of coupled modelling systems at grid points with fractional land cover, in view to ensure that shortwave fluxes to sea ice points are accounted for on every atmospheric model time step; large-scale precipitation scheme is modified to improve particle size distribution for rain and mitigate spurious feedback caused by a long-standing and explicit link between the rate of sub-grid homogenisation in the
cloud erosion parametrization and the relative humidity (Abel and Boutle, 2012); a new formulation of cloud erosion that relates the rate of mixing to the cloud fraction and reaches maximum mixing whenever the sky is half covered by clouds (Morcrette, 2012); improvements in the boundary-layer scheme to correct a significant and long-standing radiative bias within GA3.0 (Bodas-Salcedo et al., 2012). The aerosol scheme is modified to include the effect of seasonal vegetation dieback on dust emissions.

**2.2 Grid point nudging technique**

The grid-point nudging or Newtonian relaxation strategy is a form of data assimilation that helps adjust dynamical variables of free running GCMs via meteorological re-analyses. The basic idea is to constrain realistically the AGCM's solution or trajectory over or outside a prescribed domain via meteorological re-analyses to capture daily variability of key climate phenomena and examine the consequences outside or inside this domain. This technique was first applied successfully in the
mid-90s for validation of the Hamburg climate model (ECHAM) and enable the comparison of model outputs with observations over short periods of time (Jeuken et al., 1996). Since then, numerous nudged climate experiments have been carried out worldwide (Telford et al., 2008; Bielli et al., 2010; Pohl and Douville, 2011; Flaounas et al., 2012; Omrani et al., 2015; Hardiman et al., 2017). The IRCAAM (Influence Réciproque des Climats d'Afrique de l'ouest,du sud de l'Asie et du bassin Méditerranéen) project, a research initiative coordinated by the French National Research Agency (ANR), used the
grid-point nudging technique to explore remote versus regional causes of GCM' systematic errors over the West African Monsoon domain and associated teleconnections (Pohl and Douville, 2011).

The Grid-point nudging capability of the MetUM was initially developed at the University of Cambridge under the Met Office - NCAS (National Centre for Atmospheric Sciences) partnership and used for the following short-term applications: to allow



comparisons of model outputs with episodic satellites campaigns (Voulgarakis et al., 2011); to produce hindcasts over a short
period of interest, such as the eruption of Mt. Sarychev (Haywood et al., 2010); to ensure that the large scale circulation is the
same between model simulations and reanalysis; and to compare tropical convection characteristics for a given year in range
of atmospheric models, as well as numerical weather prediction (NWP) models, chemistry transport models (CTMs), and
chemistry-climate models (Russo et al., 2011).

For the MetUM, only few variables are nudged towards the 6-hourly ERA-Interim reanalysis (Dee et al., 2011) including
horizontal wind components (u and v) and potential temperature ($\theta$). This is achieved through addition of non-physical
relaxation terms to the model equations describing the rate of change for a given prognostic variable:

$$\delta X/\delta t = Fm(X) + G(Xana - X),$$

Where Fm is the rate of change in the variable due to all other factors, X is the state of the model, Xana is the reference field
toward which the model is relaxed, and G is the strength of the relaxation (Jeuken et al., 1996). Interpolation onto the MetUM
model grid is done through a bi-linear interpolation with respect to the logarithm of pressure, ln(P), from the ERA-Interim
reanalysis hybrid pressure levels to the UM hybrid height levels. The selection of parameter G is critical, such that if it is too
small nudging is ineffective, yet too large and the model becomes unstable. Here, we have chosen the "natural" value of 1/6
h−1, which corresponds to the time spacing of ERA-Interim reanalysis data. However, the relaxation parameter is set to be
weaker at the 3 lowest and 5 highest levels, to allow for a smooth model adjustment to the nudging.

**2.3 Experimental design**

Table 1 summarizes the experiments performed in this study. Three simulations are carried out continuously through 1982-
2008 (27 years) using the GA3.0 model configuration with a fixed lid at 85 km above the surface and 85 vertical levels – with
50 levels below 18 km and 35 levels above. First, GA3.0 is integrated using the N96 horizontal grid resolution i.e. 1.258
longitude by 1.8758 latitude, which corresponds roughly to 135-km in mid-latitudes. This experiment, referred to as GA3-
N96, uses daily Met Office Operational observed sea surface temperatures (SSTs) and Sea Ice Analysis (OSTIA) data (Donlon
et al., 2012), sea ice fractions, and time-varying forcings CO2 concentrations and other external forcings consistent with the
Atmospheric Model Intercomparison Project (AMIP) component of CMIP5 (Taylor et al., 2012; Ackerley et al., 2018). SSTs
are specified such that the monthly means computed from the model outputs agree with the observations (Mizielinski et al.,
2014). A second simulation, referred to as GA3-N216, is performed in all identical conditions to GA3-N96; except that the
horizontal grid resolution is enhanced to N216 (~60 km in mid-latitudes). This simulation is repeated using the N512 horizontal
grid resolution (~25km in mid-latitudes) and referred to as GA3-N512. The aim of GA3-N216, GA3-N512, and GA3-N96
experiments is to document the influence of horizontal resolution on simulated climatological features of monsoon
precipitation worldwide. One simulation is carried out continuously through 1982-2008 (27 years) using GA4.0 model



configuration, with spatial resolution and forcing conditions identical to GA3-N96. This experiment is referred to as GA4-N96
aims to explore the sensitivity of model performance to changes in the physical formulation.

Furthermore, seven sensitivity experiments are carried out continuously through 1982-2008 (27 years) with the GA3.0 model
configuration and the grid-point nudging approach – in which 6-hourly ERA-Interim reanalysis serves as reference data to
prescribe the atmosphere and ocean boundary conditions. All simulations are identical in most respects to GA3-N96. The only
difference is the area where nudging is applied at each time step to constrain the horizontal wind components (u and v) and
potential temperature (θ) at each time step, in order to exert a control on the atmospheric circulations and associated interannual
variability. These simulations are referred to as GA3-NUDG-TROPICS (Tropics), GA3-NUDG-AFRICA (Africa), GA3-
NUDG-ASM (South Asia), GA3-NUDG-EASM (East Asia), GA3-NUDG-SAP (Australia and South Asia Pacific), GA3-
NUDG-NAM (North America), GA3-NUDG-SAM (South America), and GA3-NUDG-TROPPAC (Tropical Pacific). Figure
1 illustrates the geographical areas considered for the regional nudging.

## 2.4 Validation datasets

To contextualize the MetUM Global Atmosphere configurations results, we have extended the analysis to include outputs of
20 atmospheric-only GCMs experiments, randomly selected, from the CMIP5 campaign (Table 2). These outputs were
accessed through: cmippcmdi.llnl.gov/cmip5/data_portal.html and chosen to illustrate the broad range of CMIP5 behaviour,
forced with observed SSTs and sea ice concentrations. By comparing MetUM Global Atmosphere configurations against
CMIP5 multi-models, we anticipate gaining further insights on the models' uncertainties in the monsoon regions.

The model validation process requires observational variables covering various time and space scales. But given the limited
availability of routine meteorological observations and in situ data in monsoon regions, we base the validation on gridded
datasets listed in Table 3. These include satellite estimates, gauges-based estimates, and reanalysis products including ERA-
interim dataset, which is also employed to supply initial and atmospheric boundary conditions for the grid-point nudging
strategy. ERA-Interim consists of 6-hourly estimates for three-dimensional (3D) meteorological variables and 3-hourly
estimates of surface parameters, archived with a horizontal grid resolution of approximately 79km and 60 vertical levels (with
a top at ~40km). Land precipitation are primarily assessed using the merged satellite estimates and in situ observations from
the Climate Hazards Group InfraRed Precipitation with Stations (CHIRPS) data archive (Funk et al., 2015). CHIRPS dataset
is a quasi-global (50°S–50°N, 180°E–180°W), 0.05° resolution, 1981 to present gridded precipitation time series – which
blends in more station data than other precipitation products and uses a high-resolution background climatology, providing
better estimates of precipitation means and variations (Rivera et al., 2018).



## 3 Results

### 3.1 Tropical climatology

This section assesses the ability of GA3.0 and GA4.0 to reproduce basic features of the mean tropical climate. It draws largely on the comparison between modelled quantities and corresponding observational estimates, using series of performance metrics and 'process based statistics'; developed for routine evaluation against gridded observations and reanalysis datasets of GCMs participating in fifth and sixth phase of the Coupled Model Intercomparison Project (Gleckler et al., 2008; Anav et al.,

2013; Eyring et al., 2016; Stouffer et al., 2016). These performance diagnostics are widely used to quantify similarities between models and observations and changes between different generations of models, and to explore uncertainties arising from internal variability, climate forcings and model formulations (Klein et al., 2013; James et al., 2018).

We begin by analysing GA3.0 and GA4.0 ability to reproduce contemporary climatic conditions at every grid point, and then

aggregate the results over the tropical subdomains (30°N-30°S) for the period 1984-2005 (Fig. 2). The large-scale climate responses of GA3-N96 and GA4-N96 is compared against CMIP5 models and observations, with view to portray models' behaviour relative to each other. The analysis is based on the space–time root mean-square error (RMSE) metric where individual model's RMSE is normalised by the median models' error, accounting for spatial patterns and annual cycles. Blue shading indicates the performance being better, and red shading indicates worse, than the median of all simulations. The

median rather than the mean models' error is used to prevent outliers from influencing the results. Contributing models are displayed along the horizontal axis, while large-scale climatic fields (as described in the caption) are shown on the vertical axis. Two reference datasets are used whenever possible to estimate the observational uncertainty and panel's grid squares are split into diagonals, with view to illustrate the relative error with respect to the primary (upper left triangle) and the alternate (lower right triangle) references. White triangles indicate the unavailability of an alternate reference dataset. For instance,

precipitation reference datasets include merged satellite-gauge estimates from the Global Precipitation Climatology Project (GPCP) and the Climate Hazards Group InfraRed Precipitation with Station data (CHIRPS). Wind is assessed through ERA-Interim and MERRA2 (Modern-Era Retrospective analysis for Research and Applications, Version 2) reanalyses.

In general, the performance of GA3-N96 varies across vertical levels and climate variables, and consistently outperforms that

of most CMIP5. GA3-N96 wind components compare reasonably well against ERA-Interim dataset and displays negative relative errors 10% to 20% smaller than the median at most vertical levels. Such high skills are only matched by EC-EARTH and GFDL-CM3, highlighting some benefits of using a seamless prediction system that forges weather forecasting and earth-system modelling into a single modelling framework (Hazeleger et al., 2010). At the surface, GA3-N96 also reproduces basic feature of observed precipitation and air temperature fields, and its performance matches that of best CMIP5 models including

EC-EARTH, MPI-ESM-LR, and GFDL-CM3, with relative errors of 10-20% smaller than the ensemble median model. There





simulation improves with enhancing the horizontal resolution and GA3-N216 shows RMSE values ranging between 0-10% of the ensemble median.  GA4-N96's performance displays a close similarity with GA3-N96, but there are also important differences with respect to wind components at 200 hPa, precipitation and clouds. There is a tendency for the ensemble multi-model mean and the median to agree more favourably with observations than any individual climate model. Furthermore,
choice of the reference datasets does not alter the evaluation of model performance, indicating that inter-model uncertainty is larger than observational uncertainty. This finding is consistent with the long-recognised behaviour of the coupled ocean-atmosphere ensemble mean of models participating in the IPCC AR5 assessment reports (Gleckler et al., 2008; Flato et al., 2013).

Figure 3 compares the climatological distribution of observed and modelled interannual variability (IAV) indexes, mean averages and trends for total precipitation in the tropics, relative to GPCP dataset. Here, IAV is assessed through the model variability index (MVI), as defined by Anav et al. (2013).  In regions with large/little IAV, RMSEs can often cancel out leading to overly optimistic model performance. MVI offers a solution to address these cancellation effects and provide a robust measure of model differences. Several studies suggest that RMSE can be a misleading measure for characterizing the model's
IAV and there is no reasons for model and observations to agree on the phasing of natural interannual variability, e.g. with exact timing of El Nino and La Nina events (Gleckler et al., 2008; Scherrer, 2011). While determining MVI thresholds that discriminate between ''good'' and ''poor'' model performance remains an arbitrary choice, the values of 0.5 and less are often considered an indicator of good IAV.

In the tropics, GA3-N96 and GA4-N96 large-scale behaviour is within the range of CMIP5 uncertainty and realistically capture the trends of precipitation. On the figure's x-axis, GA3-N96 falls at the right of GPCP dataset indicating  wet biases of ~17% (250 mm/year), which is comparable to biases found in CMIP5 models including HadGEM2-A, EC-EARTH, FGOALS-s2, CSIRO-Mk3-6-0, MPI-ESM-LR, MIROC5, BNU-ESM, and CESM1-CAM5. Similar biases are found in GA3-N216, GA3-N512, and GA4-N96 simulations. While no models exhibit MVI values close to the threshold of best performing simulations
(0.5), GA3-N96 tends to outperform most CMIP5 models, with values ranging between 1.32-1.48. Best performing models in terms of MVI include GA3-N512 and CESM1-CAM5, while worse results are found within FGOALS-g2, CMCC-CM and BCC-CSM1-1.

### 3.2 Global and regional monsoon features

Figure 4 shows the climatological distribution of the global monsoons precipitation intensity (GMI) index during 1984-2005
in GPCP dataset and estimates the differences between observations and model experiments. Here, GMI is considered as the global monsoon precipitation amount per unit area, and this serves as a measure of the GM strength. We follow  the approach of Lin et al. (2014a)  and define monsoons as the regions in which the annual range (AR) of precipitation exceeds 2.5 mm/day





and the local summer (May–September in the Northern Hemisphere and November–March in the Southern Hemisphere) precipitation exceeds 55% of annual total amount. AR corresponds to the difference of May–September (MJJAS) and

November–March (NDJFM) mean precipitation in the Northern Hemisphere or the difference of the NDJFM and MJJAS mean precipitation in the Southern Hemisphere.

The GPCP dataset indicates an equatorially symmetric structure of the GMI, with more intense precipitation in the tropical band and centred over core monsoon regions, and key features such as the intertropical convergence zone (ITCZ), the South

Pacific convergence zone and Indo-Pacific warm pool. These features are well captured in GA3-N96 reference experiment. GA4-N96 errors have virtually the same magnitude and spatial structure as GA3-N96, and tend to be larger over the ocean than land, with a markedly underestimate of precipitation intensity over the West African monsoon and South Asian summer monsoon region, from India to Southeast Asia and extending eastward over Indochina peninsula and Indonesia. Alternatively, precipitation is overestimated over most tropical land with positive bias values of 4 mm/day within the oceanic ITCZ and 6

mm/day over land, particularly in South Africa, South and Central America, and East Asia. In western Pacific, there is a tripole error pattern from the equator to 45N and 45S. Precipitation intensity is generally too high over the equatorial Indian Ocean, south of India. The situation improves slightly with either increasing the spatial resolution (GA3-N216 and GA3-N512) or improving the sub-grid scale parametrizations (GA4-N96). The dry bias issue over the South Asian summer monsoon region is also a striking feature of HadGEM2-A, ACCESS1-0 models, CSIRO-Mk3-6-0, GISS-E2-R, and MIROC5 experiments. By

contrast, most CMIP5 models overestimate precipitation over land in the South Asian summer monsoon region (Fig. 5).

To further assess the precipitation pattern's agreement between models and observations in the major regional monsoon areas, we use the Taylor diagram approach. This provides a concise statistical summary of the model behaviour in terms of root-mean-square difference (RMSD), pattern correlation (PCC), and ratio of variances of the model errors for seasonal mean

precipitation, with respect to reference datasets (Taylor, 2001). By "model errors" we refer to departure of the simulation from observations, assuming implicitly that observational uncertainty is smaller than model errors. For this analysis, 3 reference datasets (GPCP, CHIRPS and CRU) are selected, as well as the following 8 focus subregions corresponding to maxima of precipitation intensity in dominant monsoons areas:    West African monsoon [5°N – 20°N; 10W – 10E], South African monsoon [10°S – 30°S; 15E – 40E], North American monsoon [20°N – 30°N; 90W – 125W], South American monsoon [5°S

– 20°S; 50W – 75W],  South Asian summer monsoon [5°N – 30°N; 60E – 90E], East Asian monsoon [20°N – 50°N; 95W – 145W], Australian monsoon [10°S – 25°S; 110W – 150W], and Philippines monsoon [15°N – 10°S; 95W – 150W].

Figure 6 depicts the monsoon regions with a relatively good agreement between observation estimates and simulations. In this diagram, the PCC between any model and the reference data is related to the azimuthal angle, and the RMSD between models

and reference data is proportional to the distance between these two points. Biases, defined as the spatially averaged differences between the modelled and observed time-mean, are not shown on this diagram. In West Africa,  a region in which monsoon





precipitation is highly sensitive to the ITCZ latitudinal migration in summer and to organised mesoscale convective systems, GA3-N96 agrees favourably with the observations and model performance is comparable to that of best performing CMIP5 including MPI-ESM-LR, CMCC-CM, HadGEM2-A, and ACCESS1-0. PCC ranges from 0.9 to 0.95 and normalised RMSD

is lower than 0.5 indicating a high fidelity in reproducing the observed patterns of precipitation. GA3.0 shows also high performance over the Australian monsoon domain with PCC ranging between 0.95 and 0.99, and the ratio of model errors variances is close to 1. Conversely, GA3.0 shows medium performance over the South African and North American monsoons regions with PCC dropping to values between 0.6 and 0.8, and RSMD ranging between 0.5 and 1. There is an overall improvement of GA3.0 spatial variability with increasing the horizontal resolution. Conversely, GA3.0 and GA4.0 experience

challenges to capture monsoon precipitation patterns in the South Asia, East Asia, and South America (Fig. 7). These are regions with complex topographic features where performance is comparable to the tier of CMIP5 models and overestimate the spatial variability. GA3.0 and GA4.0 indicate relatively low spatial correlation coefficients (0.1 to 0.7) and high RMSD (1 to 1.5) values.

Since the spatial pattern of monsoon precipitation varies across regions, examining the timing and seasonality of simulated precipitation will provide further insight on the model's ability. Figure 8 shows the monthly-mean cycle of modelled and observed precipitation in three monsoon regions, computed from averaging over land points only. GA3-N96 reproduces basic features of the time-evolution of the North American monsoon precipitation including the dry season from November to May, peak in August, and the subsequent rapid decline, though this analysis can be hampered by observational uncertainty (Fig 8,

Top). There is an overestimate of precipitation during the monsoon peak and since similar results are found in in the simulations with the high spatial resolution (GA3-N216, GA3-N512) and in GA4-N96, we argue the issue is rooted within the model physics. Interestingly, HadGEM2-A and ACCESS-1-0 models agree more favourably with the observations and show improvement over GA3.0 and GA4.0 experiments.

The West African monsoon, particularly the Sahel region (10°W–10°E, 10°N–20°N), is dominated by a mono-modal precipitation cycle which begins in May, reaches a peak of 4.5 mm/day in August when the ITCZ reaches its northernmost position, and terminates in October (Fig 8, Centre). GA3-N96 captures the timing and peak of precipitation within the observational uncertainty, but also indicates a too early monsoon onset occurring in March to June. This behaviour is consistent with most CMIP5 and persists even with increasing the horizontal resolution to N216 and N512. GA3-N96 generates too little

precipitation in August, though the situation improves slightly with increasing the horizontal resolution. Over South America, GA3-N96 shows high fidelity in mimicking the monsoon precipitation's seasonal cycle (Fig. 8, bottom). The model most noticeable bias is the overestimation of precipitation during the peak season [December–March (DJFM)], compared to GPCP-SG and CHIRPS observations. This behaviour is consistent with that of most CMIP5 models.

To reflect the monsoon variability from a circulation perspective, we assess the mean seasonal cycle of Webster and Yang

index (WYI) in models and reanalysis, computed as a time-mean zonal wind (U) shear between 850 and 200 hPa, U850−U200,



averaged over south Asia from the equator to 20°N and from 40° to 110°E (Fig. 9). The WYI provides a first-order measure of the South Asian Summer Monsoon strength and represents the combined effect of the low-level (850 hPa) westerly jet and upper-level (200 hPa) easterly jet, two important features of in this region. Generally, GA3-N96 reproduces well the timing and peak of monsoon strength and shows consistency with most CMIP5 models.

### 3.3 Impact of remote versus local atmospheric circulations errors

To disentangle the impact of remote versus local atmospheric circulation errors on the GA3.0 and GA4.0 performance, we focus on the Asian Summer Monsoon, a system for which the complexity causes great challenges to many AGCMs in simulating the associated climatological seasonal means and annual cycles of precipitation (Sperber et al., 2013; Singh, 2015; Cherchi et al., 2016; Annamalai et al., 2017; Anand et al., 2018).

Figure 10 illustrates the mean climatological wind at 850 hPa during the JJAS season, in the Asian summer monsoon (ASM) domain, for GA3.0, GA4.0 and some selected atmosphere only models from the CMIP5 dataset. Key characteristics of the low levels' circulations are represented with the same consistency within ERA-Interim, MERRA2, CFSR, and JRA reanalyses including the location of the Somali jet and the turning of the westerly flow northward into the Bay of Bengal. GA3-N96 control experiment captures reasonably well wind directions in comparison to various re-analysis. The model most striking bias is the weak strength of low-level Somali jet that carries moisture from over ocean to the Indian land region, a behaviour that persists with increasing the MetUM horizontal resolution to N216 and N512. This systematic error is also apparent in CSIRO-Mk3-6-0 and ACCESS1-0, but contrasts with EC-EARTH which indicates substantial improvement in the monsoonal flow in this region. The GA3.0 best results are obtained with the atmospheric circulations realistically constrained over the tropics and West African monsoon and Asian Summer monsoon domains, whereby spatial correlation coefficients with the ERA-Interim data range from 0.96 to 0.97 (Figure 11). Conversely, nudging of atmospheric circulations over the tropical Pacific and other remote monsoon domains show little positive influence on the Somali Jet's strength. This suggests a dominant role for local/regional rather than remote atmospheric circulations errors in GA3.0 simulations of the ASM domain. GA4.0 shows a similar tendency to underestimate the strength of the ASM low level circulations.

Figure 12 displays the observed and simulated climatological distribution of global monsoon areas for the GA3-N96 and all nudging experiments, computed using GPCP satellite estimates, considered as the observed ''ground truth''. The observation highlights Northern Hemisphere (NH) and the Southern Hemisphere (SH) monsoon domains, as well as few oceanic monsoon spots such as in South China, from the East China Sea to south of Japan. The SH monsoon consists of the South African, Australian and South American monsoons. The NH monsoon includes the South Asian, East Asian-western North Pacific, West African, and North American monsoons. There is also a minor region in the central South Pacific with clear alternation





between wet summers and dry winters, but without land–ocean thermal contrasts, and which is not regarded as a monsoon region in this analysis.

Globally, GA3-N96 reproduces the extent of most domains experiencing dry-wet alteration in the tropical Asia, Australia, Africa, and the Indian Ocean. But the model falls short in simulating the South Asian monsoon domain, with the ratio of local summer precipitation to annual rainfall being less than 0.55. The failure over the ASM domain persists with increasing the model horizontal resolution (GA3-N216 and GA3-N512) and enhancing the parametrizations of sub-grid scale processes (GA4-N96), and it is common to CSIRO-Mk3-6-0 and ACCESS1-0 – two climate models sharing some dynamical and

physical components with the MetUM Global Atmosphere configurations (not shown). Meanwhile, constraining the atmospheric circulations toward ERA-Interim reanalysis values leads to substantial improvement, especially if the grid-point nudging technique is applied over the tropical and ASM domains (GA3-NUDG-TROPICS and GA3-NUDG-ASM).

Figure 13 shows that the observed and GA3-N96 core South Asian summer precipitation occurs over land during June-July-

August-September (JJAS). The observations show that the monsoon onset occurs with an abrupt increase in precipitation during early June, peaking during July and followed by a retreat at a relatively slower rate as compared to the onset. GA3-N96 captures the timing of the climatological annual cycle of precipitation, but there is a significantly dry bias with respect to the total amount. The annual cycle improves considerably with the ASM and tropical atmospheric circulations realistically constrained. We hypothesize that poor simulations of precipitation over South Asia by GA3.0 and GA4.0 are attributable

partly to atmospheric circulations errors over and excessive precipitation over the southwest equatorial Indian Ocean, rather than to tropical atmospheric responses of varying forcing fields, such as SST over the Arabian Sea, aerosols, and growing greenhouse gas emissions as suggested by several studies for some CMIP5 models (Hsu, 2016; Kitoh et al., 2013; Levine et al., 2013). But this needs further examination, which is beyond the scope of present paper.

## 4 Conclusion

Based on the Coupled Model Intercomparison Project/Atmospheric Model Intercomparison Project AMIP framework, which uses observational SST and sea ice to drive AGCMs simulations, we have evaluated the features of regional and global monsoon hydroclimates in the MetUM Global Atmospheric configurations GA3.0 and GA4.0 – two configurations developed for seamless use across climate and weather time scales. GA4.0 uses dynamical and physical cores that are largely identical to GA3.0 but includes some improvements in the physics. GA3.0 is first integrated to explore the influence of

resolution on model performance, using separately three horizontal resolutions: a low (~135-km in mid-latitudes), medium (~60 km in mid-latitudes) and high (~25km in mid-latitudes). Second, GA4.0 is integrated following the exact condition of GA3.0 reference experiment to explore the impact of changes in the parametrization of sub-grid scale processes. Third, we further investigate, for the first time, the impact of remote versus local atmospheric circulation errors on the GA3.0



performance in monsoon regions using the "grid-point nudging" or "Newton relaxation" technique - whereby the AGCM's atmospheric circulation fields are fully constrained by reanalysis over or outside a prescribed domain in view to examine the consequences outside or inside this domain, and to provide the most realistic representations of the atmosphere at a given time. To provide further context, the results are compared against 20 atmospheric-only GCMs simulations from the CMIP5 dataset.

The results indicate that GA3.0 reproduces realistically climatological features of the monsoon climate worldwide including the geographical domains, precipitation intensity and annual cycle. GA3.0 also compares well against the best available CMIP5 models and agrees more favourably with the observations over the West African, South African, North American and Australian monsoons domains. The most striking model failure is the significant dry bias over the South Asian summer monsoon domain. There is clear benefit in increasing GA3.0 horizontal resolution, particularly at N216 (~60 km in mid-latitudes) and N512 (~25km in mid-latitudes), although this does not alter the broad structure of model systematic errors in monsoon regions. GA4.0's behaviour and structural errors displays close similarity with GA3.0, but there are some important differences with respect to precipitation, clouds and wind fields in the upper atmosphere. At large-scale, GA3.0 and GA4.0 configurations outperform many of the CMIP5 models with respect to interannual variability of tropical precipitation including timing and seasonality.

The GA3.0 and GA4.0 most striking biases is a failure to capture precipitation intensities over the South Asian summer monsoon (ASM) domain and the magnitude of low-level Somali jet; that carries moisture from over ocean to the Indian land region. Precipitation intensity is underestimated over the Philippines Sea, Indonesia, and part of the Indo-China Peninsula. Conversely, precipitation intensity is generally too high over the equatorial Indian Ocean, south of India. This feature is also common to HadGEM2-A, ACCESS1-0, INMCM4, GISS-E2-R, MIROC5, and CSIRO-Mk3-6-0 experiment. The analysis of remote versus local atmospheric circulations impacts indicates that poor simulations over South Asia in GA3.0 and GA4.0 are connected with the Somali low level jet and attributable partly to local atmospheric circulations errors and excessive precipitation over the southwest equatorial Indian Ocean, rather than to remote tropical atmospheric responses of varying forcing fields, such as SST, in other remote monsoon regions. To increase confidence and robustness in monsoon projections and climate mitigation strategies, we stress the importance of carefully considering the ability of climate models to represent interactions across different regional monsoon systems for future projections of the hydrological cycle.

**Author contributions**

W. Moufouma-Okia designed, run and analysed the experiments, processed the multiple sources of observations and model data, and wrote the manuscript. D.A. Vondou assisted in the analysis and produced Figure 6 and 7 and commented the manuscript and contributed with ideas and suggestions. R. Jones motivated the investigation, supervised its development and provided modelling infrastructure and resources.





**Acknowledgments, Samples, and Data**

The authors thank the team of ESMvaltool free visualization software which was used to analyse observational and model datasets (Eyring et al. 2016a). CMIP5 and outputs can be directly accessed through the following website https://esgf-node.llnl.gov/projects/cmip5/. Observations and reanalyses can be accessed through https://esgf-node.llnl.gov/projects/obs4mips/. All analysed GA3 and GA4 model outputs are available on request to the corresponding author through a secured ftp server.

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

**Tables**

Table 1: GA3.0 and GA4.0 Global Atmosphere configurations experiments performed in this study. Grid-points nudged experiments have been carried out using the GA3.0 model configuration with N96 horizontal resolution i.e. 1.875° x 1.25° (~135 km in mid-latitudes) and 85 vertical levels. Experiments performed with N216 and N512 horizontal resolutions correspond respectively to ~60 km and ~25 km in mid-latitudes.

| Experiment name | Experiment details | Integration period |
|---|---|---|
| GA3-N96 | GA3.0 run with N96 resolution i.e. 1.875° x 1.25° (~135 km) | 1982-2008 |
| GA3-N216 | GA3.0 run with N216 resolution i.e. 0.83° x 0.56° (~60 km) | 1982-2008 |
| GA3-N512 | GA3.0 run with N512 resolution i.e. 0.35° x 0.234° (~25 km) | 1982-2008 |
| GA4-N96 | GA3.0 run with N96 resolution i.e. 1.875° x 1.25° (~135 km) | 1982-2008 |
| GA3-NUDG-TROPICS | run with grid-points nudging over the Tropics [35°N – 35°S; 180W – 180E] | 1990-2008 |
| GA3-NUDG-AFRICA | run with grid-point nudging over Africa [42°N – 42°S; 25W – 60E] | 1990-2008 |
| GA3-NUDG-ASM | run with grid-point nudging over Asia Summer monsoon domain [10°S – 40°N; 60E – 120E] | 1990-2008 |
| GA3-NUDG-EASM | run with grid-point nudging over East Asia Summer monsoon domain [10°N – 70°N; 90E – 160E] | 1990-2008 |
| GA3-NUDG-SAP | run with grid-point nudging over Australia and South Asia Pacific [10°N – 60°S; 90E – 180E] | 1990-2008 |
| GA3-NUDG-NAM | run with grid-point nudging over North America [15°N – 65°N; 50W – 150W] | 1990-2008 |





| GA3-NUDG-SAM | run with grid-point nudging over South America [15°N – 65°S; 30W – 90W] | 1990-2008 |
|---|---|---|
| GA3-NUDG-TROPPAC | run with grid-point nudging over Tropical Pacific [35°N – 35°S; 150E – 250E] | 1990-2008 |
| GA3-NUDG-GLOBAL | Run with grid-point nudging over the entire planet | 1990-2008 |

860 Table 2: Atmospheric only (AMIP) GCMs participating in the Coupled Model Intercomparison Project Phase 5 (CMIP5) and included for comparison with the MetUM Global Atmosphere model configuration. These climate simulations are available at monthly time scale through the period 1979-2005.

| Model | Experiment name | Horizontal resolution in latitude by longitude (km), vertical levels | Country of the Modelling centre | References |
|---|---|---|---|---|
| HadGEM2 | HadGEM2-A | 1.875° x 1.25° (~135 km), L38 | UK | (Jones et al. 2011) |
| ACCESS1-0 | ACCESS1-0 | 1.875° x 1.25° (~135 km), L38 | Australia | (Bi et al. 2013) |
| CSIRO-Mk3-6-0 | CSIRO-Mk3-6-0 | 1.8653° x 1.875°, L18 | Australia | (Jeffrey et al. 2013) |
| IPSL-CM5A-LR | IPSL-CM5A-LR | 1.8947° x 3.75°, L39 | France | (Dufresne et al. 2013) |
| INM-CM4 | inmcm4 | 1.5° x 2°, L21 | Russia | (Volodin et al. 2010) |
| MIROC5 | MIROC5 | 1.4008° x 1.40625°, L40 | Japan | (Watanabe et al. 2010) |
| MPI-ESM-LR | MPI-ESM-LR | 1.8653° x 1.875°, L47 | Germany | (Giorgetta et al. 2013) |
| NorESM1-M | NorESM1-M | 1.8947° x 2.5°, L26 | Norway | (Bentsen and Roelandt 2012) |
| BCC-CSM1-1 | BCC-CSM1-1 | 2.7906° x 2.8125°, L26 | China | (Xiao-Ge et al. 2013) |





| CanAM4 | CanAM4 | 2.8125° x 2.7906°, L35 | Canada | (Von Salzen et al. 2013) |
|---|---|---|---|---|
| CCSM4 | CCSM4 | 0.9424 x 1.25°, L26 | USA | (Gent et al. 2011) |
| CESM1-CAM5 | CESM1-CAM5 | 2.5°x 0.9424°, L26 | USA | (Kay et al. 2015) |
| CMCC-CM | CMCC-CM | 3.7111° x 3.75°, L31 | Italy | (Scoccimarro et al. 2011) |
| CNRM-CM5 | CNRM-CM5 | 1.4008° x 1.40625°, L31 | France | (Voldoire et al. 2013) |
| EC-EARTH | EC-EARTH | 1.1215° x 1.125°, L62 | Netherlands | (Hazeleger et al. 2010) |
| FGOALS-g2 | FGOALS-g2 | 2.7906° x 2.8125°, L26 | China | (Li et al. 2014) |
| FGOALS-s2 | FGOALS-s2 | 1.6590° x 2.8165°, L26 | China | (Bao et al. 2013) |
| GFDL-CM3 | GFDL-CM3 | 2° x 2.5°, L48 | USA | (Griffies et al. 2011) |
| BNU-ESM | BNU-ESM | 2.7906° x 2.8125°, L26 | China | (Ji et al. 2014) |
| GISS-E2-R | GISS-E2-R | 2° x 2.5°, L40 | USA | (Miller et al. 2014) |

865 Table 3: Datasets used for validation. For each dataset, the acronym used hereafter, the field used and the horizontal resolution are shown.

| Quantity | Variable Name | Observations | Horizontal resolution | Reference |
|---|---|---|---|---|
| Surface (2 m) Air Temperature (°C) | Tas (2 m) | ERA-Interim or IFS-Cy31r2 | ~79 km | (Dee et al. 2011) |
| | | NCEP-CFSR | ~38 km | (Saha et al. 2010) |
| | | MERRA | 0.5 ° x 0.667 ° | (Rienecker et al. 2011) |
| | | CRU TS 3.10 | 0.5°x0.5° (~50km) | (Harris et al. 2014) |
| | | UDEL | 0.5°x0.5° (~50km) | (Legates and Willmott 1990) |
| Total precipitation (mm day–1) | Pr | GPCP | 1°x1° (~100km) | (Adler et al. 2003) |
| | | CHIRPS | 0.05°x0.05° (~5km) | (Funk et al. 2015) |



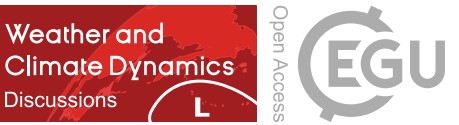

|  |  | TRMM 3B43 | 0.25°x0.25° (~25km) | (Huffman et al. 2007) |
|---|---|---|---|---|
|  |  | CRU TS 3.10 | 0.5°x0.5° (~50km) | (Harris et al. 2014) |
|  |  | UDEL | 0.5°x0.5° (~50km) | (Legates and Willmott 1990) |
| TOA longwave radiation (W m–2) | Rlut | CERES EBAF 2.6 | 1°x1° (~100km) | (Loeb et al. 2018) |
| Total cloud fraction (%) | CLT | CRU TS 3.10 | 0.5°x0.5° (~50km) | (Harris et al. 2014) |
| Zonal mean wind (m s–1) | Ua (200, 600, 850 hPa) | ERA-Interim or IFS-Cy31r2 | ~79 km | (Dee et al. 2011) |
|  |  | NCEP-CFSR | ~38 km | (Saha et al. 2010) |
|  |  | MERRA | 0.5 ° x 0.667 ° | (Rienecker et al. 2011) |
| Meridional mean wind (m s–1) | Va (200, 600, 850 hPa) | ERA-Interim or IFS-Cy31r2 | ~79 km | (Dee et al. 2011) |
|  |  | NCEP-CFSR | ~38 km | (Saha et al. 2010) |
|  |  | MERRA | 0.5 ° x 0.667 ° | (Rienecker et al. 2011) |

**Figures**



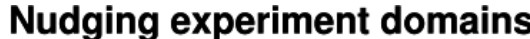

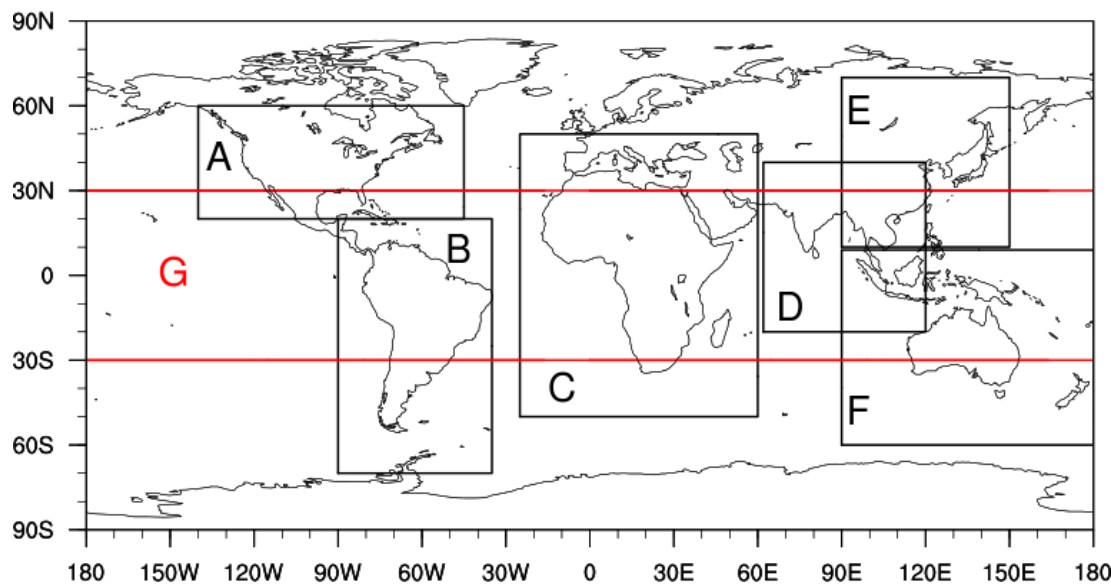

Figure 1: Domains used to relax GA3.0 prognostic fields [horizontal wind components (u and v) and potential temperature (θ)] towards the 6-hourly ERA-Interim reanalysis. These include: (box A) North America [15°N – 65°N; 50W – 150W], (box B) South America [15°N – 65°S; 30W – 90W], (box C) Africa [42°N – 42°S; 25W – 60E], (box D) Asia Summer monsoon [10°S – 40°N; 60E – 120E], (box E) East Asia Summer monsoon domain [10°N – 70°N; 90E – 160E], (box F) Australia and South Asia Pacific [10°N – 60°S; 90E – 180E], (box G) Tropics [30°N – 30°S; 180W – 180E], and (box not shown) Tropical Pacific [35°N – 35°S; 150E – 250E].





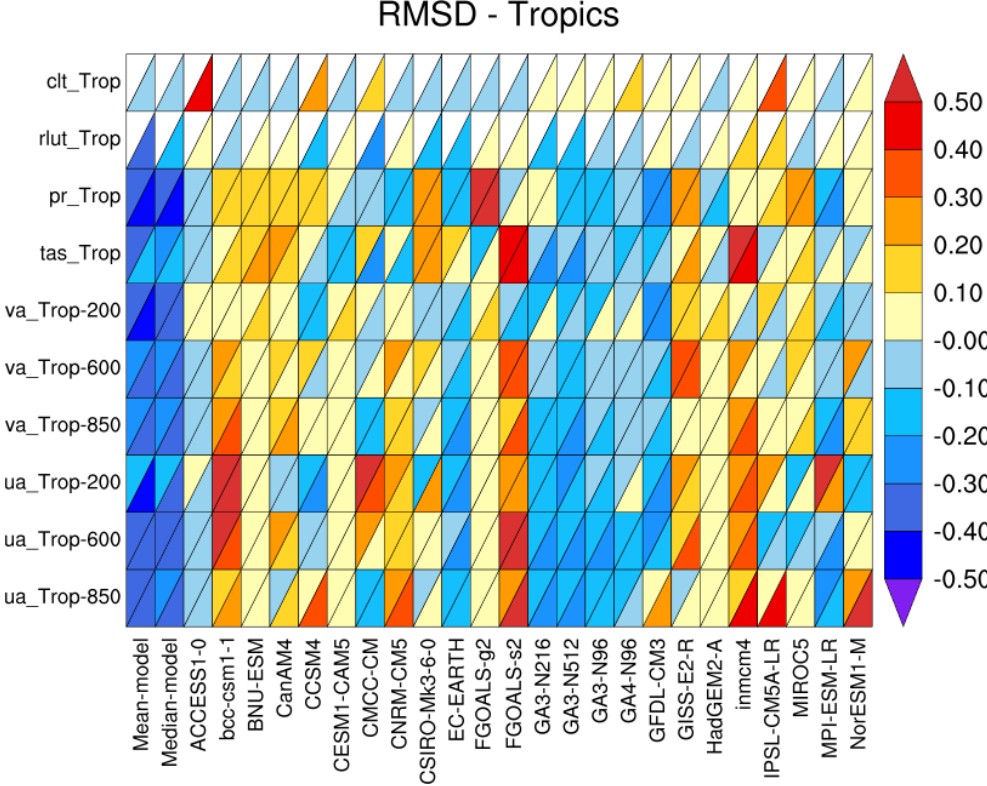

Figure 2: Relative error of model performance in the tropics, based on mean seasonal-cycle climatology (1984-2005). Rows and columns depict individual climate variables and models, respectively including GA3-N96, GA3-N216, GA3-N512, GA4-N96, and 20 available CMIP5. Results are normalized by the median error of all models. No colour indicates that model results are unavailable. The diagonal split of a grid square depicts the relative error with respect to the default reference dataset (upper left triangle) and the alternative dataset (lower right triangle).



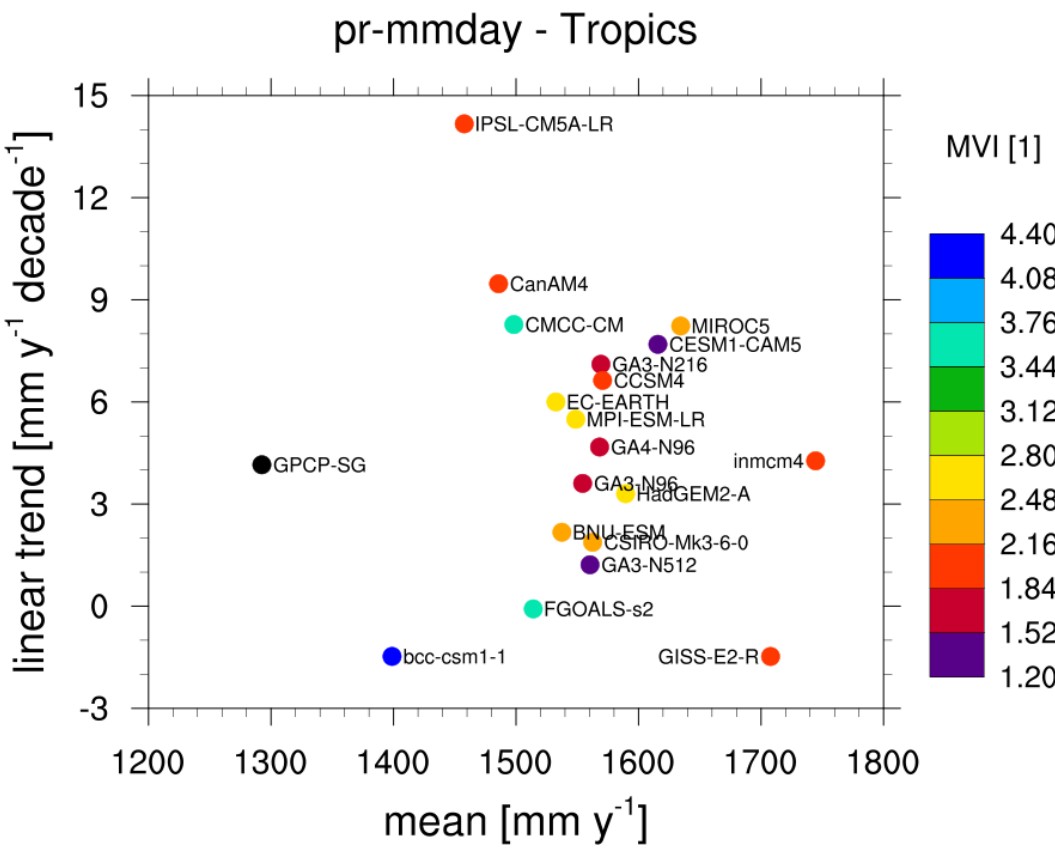

Figure 3: Model variability index (MVI) and linear trend of monthly precipitation in GA3-N96, GA3-N216, GA3-N512, GA4-
890 N96, and 20 available CMIP5 models for the tropical domain (30°N-30°S) – with respect to GPCP-SG dataset. The diagram
shows multiyear and global scale precipitation in x axis during the period 1984–2005, linear trend in y axis, and MVI in the
colour scale.



Figure 4: Monsoon precipitation intensity (mm/day) in GPCP dataset and systematic errors in GA3-N96, GA3-N216, GA3-N512, GA4-N96, HadGEM2-A, ACCESS1-0, and CSIRO-Mk3-6-0 models for 1984-2005.



Figure 5: Monsoon precipitation intensity (mm/day) in GPCP dataset and systematic errors in GISS-E2-R, IPSL-CM5A-LR, INM-CM4, MIROC5, MPI-ESM-LR, NorESM1-M, BCC-CSM1-1 models for 1984-2005.

905



Figure 6: Taylor diagrams displaying normalized pattern statistics of monthly precipitation in 1984-2005 over the land part of
West African monsoon (top left), South African monsoon (top right), North American monsoon (bottom left), and South
American monsoon (bottom right) cores. Plots compare the experiments GA3-N96, GA3-N216, GA3-N512 against 20
members of CMIP5 models forced with observed SSTs, and 3 gridded observational datasets (CRU, CHIRPS, and GPCP).
The GPCC gauges-based dataset serves as reference for the analysis.



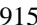

915

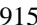

Figure 7: Similar to Fig. 6, but over South Asian monsoon (top left), Australian monsoon (top right), East Asian monsoon (bottom left right), and Philippines monsoon (bottom right) land domains.

920

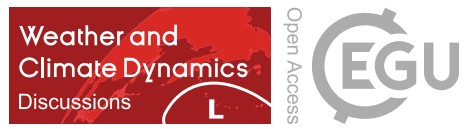





Figure 8: Annual cycle of monthly-mean precipitation in 1984-2005 for the North American monsoon (top), the West African monsoon (centre), and South American monsoon (bottom) regions as defined in Fig. 6. The data include GA3.0, GA4.0 and CMIP5 experiments, as well as

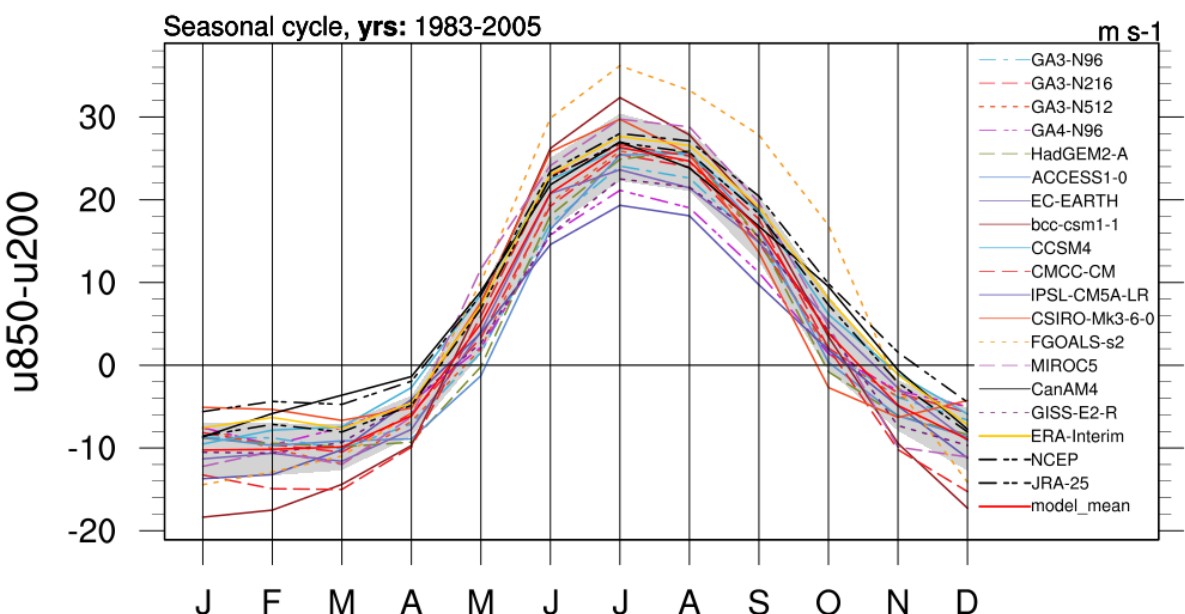

Figure 9: Annual cycle of Webster Yang monsoon circulation Index (WYI) in 1984-2005 for the South Asian Summer Monsoon. Data include GA3.0, GA4.0 and CMIP5 experiments, as well as reanalyses (ERA-Interim, JRA-25). Shade represent the interquartile model ensemble spread (range between the 25th and 75th quantiles).



Figure 10: The Asian summer monsoon mean June-September (JJAS) wind speed (m/s) at 850-hPa for 1984-2005, in MetUM Global Atmosphere configurations (GA3-N96, GA3-N216, GA3-N512, GA4-N96), CMIP5 experiments and reanalyses (MERRA2, CFSR, JRA-25, and ERA-Interim). Mean value and spatial correlation coefficient are also indicated.



Figure 11: The Asian summer monsoon mean June-September (JJAS) wind speed (m/s) at 850-hPa for 1984-2005, in the nudging experiments GA3-N96, GA3-NUDG-GLOBAL, GA3-NUDG-TROPICS, GA3-NUDG-TROPPAC, GA3-NUDG-AFRICA, GA3-NUDG-ASM, GA3-NUDG-EASM, GA3-NUDG-SAP, GA3-NUDG-NAM, GA3-NUDG-SAM, GA4-N96, and ERA-Interim reanalysis. Mean value and spatial correlation coefficient are also indicated.



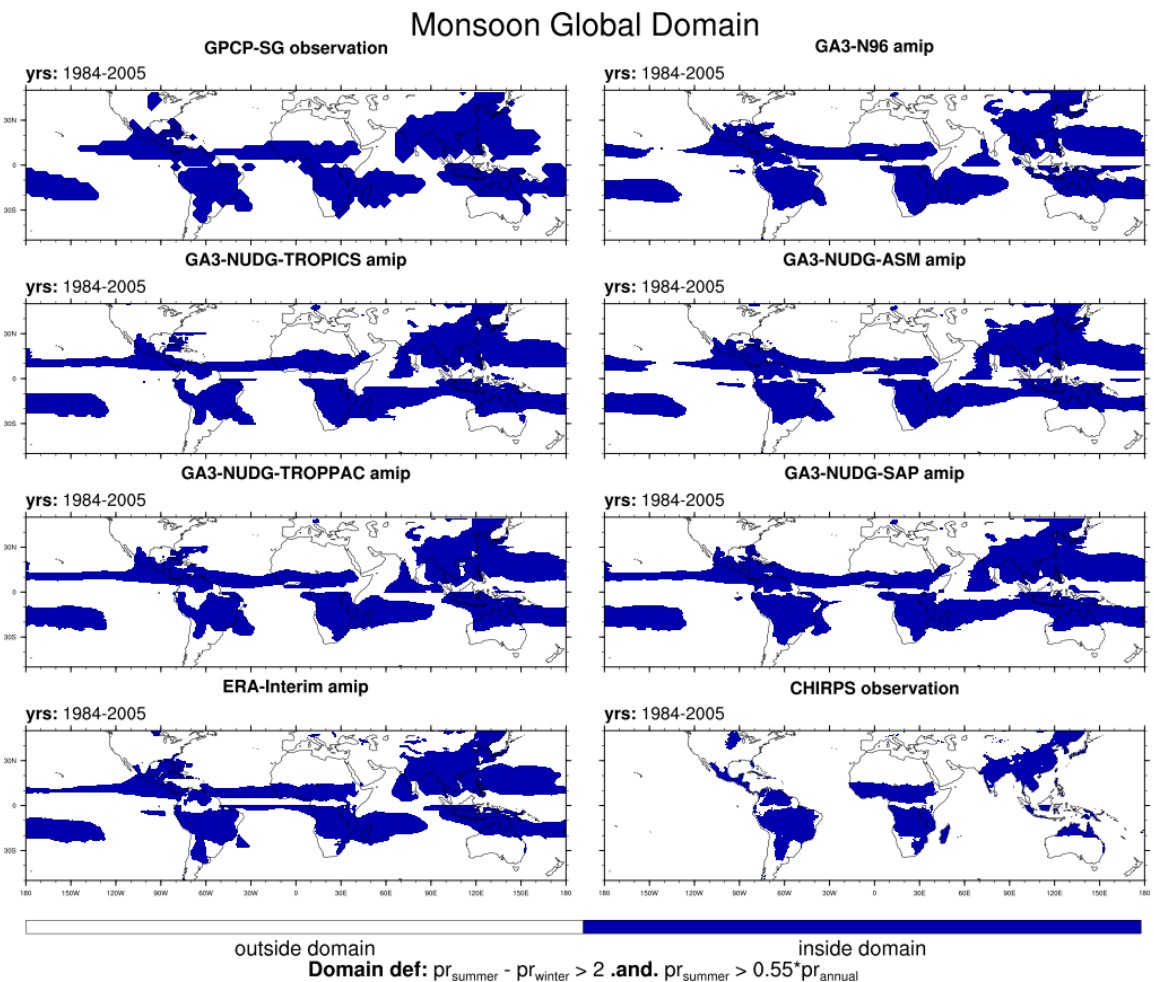

Figure 12: Climatological distribution global monsoon domain (shading) during 1984-2005 in the nudging experiments GA3-N96, GA3-NUDG-GLOBAL, GA3-NUDG-TROPICS, GA3-NUDG-TROPPAC, GA3-NUDG-AFRICA, GA3-NUDG-ASM, GA3-NUDG-EASM, GA3-NUDG-SAP, GA3-NUDG-NAM, GA3-NUDG-SAM, GA4-N96, and ERA-Interim reanalysis. The observations are based on GPCP and CHIRPS datasets.

950



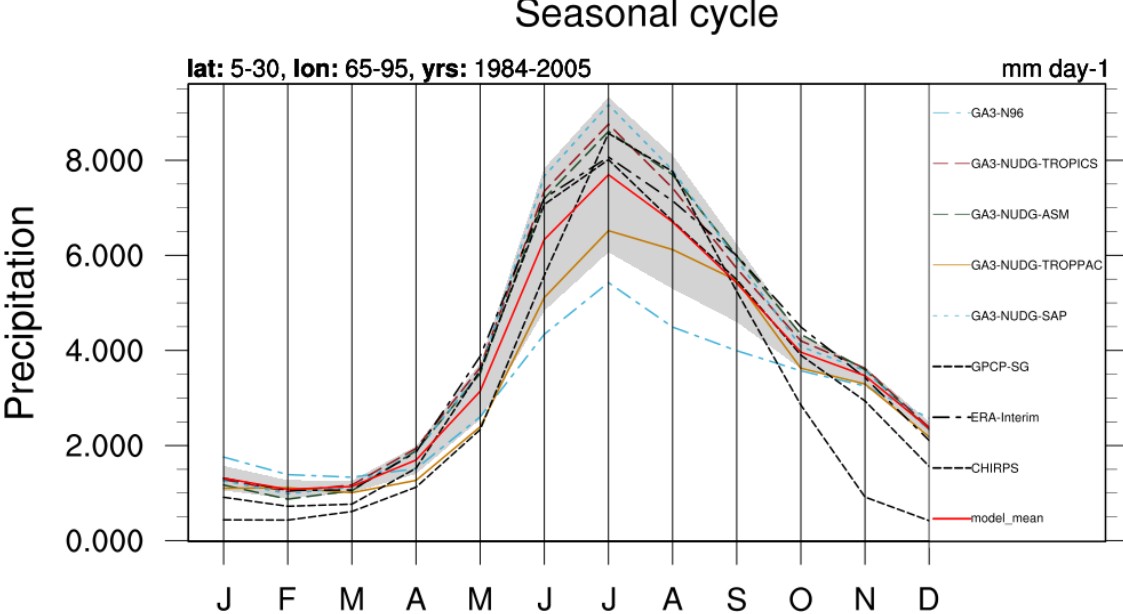

955

Figure 13: Climatological Annual cycle of monthly precipitation during 1984-2005 in CHIRPS and GPCP observations, ERA-Interim, and in the grid-point nudging experiments: GA3-N96, GA3-NUDG-GLOBAL, GA3-NUDG-TROPICS, GA3-NUDG-TROPPAC, GA3-NUDG-AFRICA, GA3-NUDG-ASM, GA3-NUDG-SAP. Shade represent the interquartile model ensemble spread (range between the 25th and 75th quantiles)