# Peer review of "The monsoon hydroclimates in HadGEM3 model configurations GA3.0 and GA4.0: Impact of remote versus local circulations errors and horizontal resolution"

_Weather and Climate Dynamics, 2020_

## Referee Comment (RC1) · Anonymous Referee #1 · 28 Oct 2020

The article investigates the sensitivity of monsoon systems to different model configurations. Analysis includes two versions of the HadGEM3model, the use of different horizontal resolutions and the application of grid-nudging to specific areas where model outputs are relaxed towards reanalyses. I appreciated the plethora of datasets and the approaches that the authors employed to reach their results. Nevertheless, I have three major concerns about this manuscript.

1) My first concern goes on the use of English and the general presentation of the results. A selection of minor comments below point to that direction. In fact, several

sentences even use familiar language coming at the expense of technical language.

2) I found that the intercomparison between models, HadGEM3 and observations remains to a rather superficial level. Most of the text presents the content of the figures in a rather descriptive way. This makes the text less attractive and my impression is that section 3 suffers from the lack of interpretation of the results with regards to the paper objectives. Actually this brings me to the third major concern.

3) My general impression is that the manuscript lacks of focus. Using such an impressive amount of datasets, applied to such a high range of monsoon systems is by all means very welcome. However, this causes the text to attain a hasty character that tries to cover all aspects of a multidimensional analysis. Therefore, it is hard to follow the manuscript. I strongly suggest to the authors to:

i) Be very precise, even by numbering, the objectives in the introduction: The paper content seems to lean towards the model validation aspects, but it also aims "to disentangle the impact of remote versus local atmospheric circulation error". Please provide the objectives and give equal weight to the analysis of each.

ii) Present all diagnostics and methods in section 2 and link them to the objectives: For instance, several indexes and datasets are first introduced in section 3.

iii) Present the results in a consistent way, e.g. all diagnostics are applied to all monsoon systems. For instance, it is characteristic that not all of the monsoon areas are treated equally. If certain areas need special attention, e.g. Asian or African monsoon, then this could be potentially justified by your results and thus more focused analysis could come in a subsequent section.

My suggestion is either to perform a major revision, or to withdraw the article and resubmit it after adapting the analysis and the text to more precise objectives.

Minor comments

Title: At first glance, at least for me, it was not clear if the paper focuses on a specific

monsoon region, e.g.: "Hydorclimates of monsoon regions in....".

Abstract: I believe that the abstract would benefit from a more structured presentation. For instance, in line 13 (This work examines..) you could be more precise on the evaluated variables and regions. Then, in the following sentences you could clearly state the methods that you used (resolutions, nudging, observations...) and finally you could present the most important results and conclusions.

Line 16: "Model outputs" instead of "Results"?

Lines 31-33: This phrase seems to be out of context. Maybe it suits better to the beginning of the Abstract.

Line 43-44: "while there is a decreasing trend of lower-troposphere monsoon circulations", this phrase is a bit confusing.

Lines 45: "monsoon coverage" is meant spatial extent of the ITCZ? Could you please clarify. If indeed this refers to the spatial coverage as in line 43, then what is meant by stronger?

Lines 60: " Monsoons are affected by..." The systems/factors mentioned in this sentence have positive and negative feedbacks in precipitation, inland propagation of the rainbands, seasonal rainfall amounts etc. It would be useful to develop these aspects to the extent that the reader acquires a deeper insight into the complexity of the monsoon systems and the challenge faced by the models to properly represent them.

Line 85: Precipitation amounts?

Lines 87-94: A series of phrases refer to interactions, linkages, changes, connections. Could you please further develop each phrase to be more precise, e.g. what is meant by "volcanic forcing-induced-intensification", which hemisphere? Why is ENSO important for the Australian monsoon?

Line 94: Please remove "climate usually" -> "The West African Monsoon interacts..."

Line 96: This rather long paragraph could benefit from a concluding phrase.

Lines 98-112: I found this part to be of very general nature and less attractive than the following paragraph. I suggest to either omit it, or to make it more relevant to the article's objectives and content.

Lines 131-133 & 135-138. These phrases are complex and hard to understand.

Lines 128- 141: Overall this paragraph lacks of being precise on the paper objectives. Reading the paper up to here, I have a general understanding of the motivation and methods. On the other hand, I am not sure about the paper's contribution to the state of the art and about the methods used to reach them.

Lines 216-220: This part should be moved to the introduction when stating the objectives.

Line 253: Why the quotes to "process based statistics"?

Line 272: Please include all datasets used in this study to section 2.4 (e.g. MERRA2 is missing).

Lines 290-292: Please be more informative about IAV and MVI and about their importance for your study.

Line 308: This title is rather too general. Could you please be more precise to the content of the section?

Line 312: AR definition should come before this line. Is it possible to show the monsoon regions in Fig. 4?

Line 327: "The situation improves..." is familiar language. This also stands for most of this paragraph. Please be more specific by quantifying the performed comparisons.

Line 336: replace "3" by "three" and "8" by "eight".

Line 342: Maybe you could skip the paragraph break.

Line 381: As previously, description of the index should come earlier, in the previous sentence.

Figures 4,5 and 12: The geographical coordinates and colorbars use very small font sizes.

Line 381: As previously, description of the index should come earlier, in the previous sentence.

Figures 4,5 and 12: The geographical coordinates and colorbars use very small font sizes.

[Figure]

---

## Referee Comment (RC2) · Anonymous Referee #2 · 15 Nov 2020

Moufouma-Okia et al examined the fidelity of the Met Office Unified Model (MetUM) in simulating the global monsoons climatological features. They have considered the MetUM third and fourth generations Global Atmosphere models and compared the results against multiple observational datasets as well as several atmospheric-only GCMs simulations from the CMIP5 experiments. The improved understanding of the GCM performance is important for studying the variability and projections of global monsoon changes. This is an interesting study on how the remote versus local circulations errors; and the horizontal resolutions of the model influencing the monsoon circulation

and related precipitation. However, I would like the authors to address some of my concerns.

Major Comments:

One of the major conclusions given in the abstract and conclusions about the poor simulations of Asian summer monsoon (ASM) by the model which was attributed to excessive precipitation over the southwest equatorial Indian Ocean, rather than to remote tropical atmospheric responses of varying forcing fields, such as SST over the Arabian Sea, aerosols, and growing greenhouse gas emissions. However, this statement was not supported by any kind of analysis. There is no such analysis for the SST over the Arabian Sea, aerosols, and growing greenhouse gas emissions are presented in this manuscript. Also, how the excessive precipitation over the southwest equatorial Indian Ocean contributed to poor simulations of the ASM are not explained.

It is interesting to note that HadGEM3 performance improves significantly over ASM with atmospheric circulations constrained realistically over the tropics, West African and Asian summer monsoon domains. However, it is not clear how exactly that contributed to such performance improvement. Is this due to the improvement of regional low-level circulation or due to the improvement in simulating the vertical wind shears? Are there any differences in the local circulations observed for ASM when atmospheric circulations constrained over the tropics, West African and Asian summer monsoon domains?

Again, you state that the increasing spatial resolution or improving the sub-grid scale parameterizations improves the model simulations (L327-330). This is not always true as in certain cases in your results the simulation doesn't improve even with a higher spatial resolution. The authors need to elaborate more on this.

Minor comments:

L222: Here it is mentioned that the nudging experiments are carried out through 1982-

2008, but you have shown in Table 1 that the integration period for the nudging experiments are 1990-2008. Which is correct?

L82-83: "South African monsoon" – is this monsoon over the country South Africa or over the South African region?

L260: why do you focus the results only for the period 1984-2005 although your simulation periods are for the period 1982-2008?

Fig. 2: Mean and Median of the models - is this the mean for all the CMIP5 models only or the mean for both CMIP5 models and HadGEM3 experiments

Fig. 5: Are these results statistically significant?

L346-347: "In west Africa, a region. . .." - give a reference.

Fig. 8: What does the grey shade indicate?

L360-370: Why did you focus only three monsoon regions here?

L453: "There is clear benefit in increasing horizontal resolution" – this is not always true. Please explain what exactly improved? Whether the increased horizontal helped improving the simulated precipitation patterns, intensity, etc.

Fig. 9 and related results discusses only about the differences in the circulation characteristics, but how do such differences influence the simulated precipitation strength in the models are not discussed.

Is the regional monsoon precipitation (e.g. seasonal cycle) in this study calculated only over land or both over land/ocean?

Fig. 12: Domain Def: Psummer – Pwinter > 2. Is this 2.5 instead of 2 as per L78?

---

## Author Comment (AC1) · 31 Dec 2020

Dear Reviewer RC1, We are grateful for your careful thoughts and comments and pleased to submit our proposed strategy for revising the manuscript entitled "The monsoon hydroclimates in HadGEM3 model configurations GA3.0 and GA4.0: Impact of remote versus local circulations errors and horizontal resolution".

The aim is to revise the manuscript considering the different comments you have raised in the text below:

[Figure]

The article investigates the sensitivity of monsoon systems to different model configurations. Analysis includes two versions of the HadGEM3model, the use of different horizontal resolutions and the application of grid-nudging to specific areas where model outputs are relaxed towards reanalyses. I appreciated the plethora of datasets and the approaches that the authors employed to reach their results. Nevertheless, I have three major concerns about this manuscript.

1) My first concern goes on the use of English and the general presentation of the results. A selection of minor comments below point to that direction. In fact, several sentences even use familiar language coming at the expense of technical language.

Answer: We agree with the comments and will amend both the text and general presentation to include more technical language.

2) I found that the intercomparison between models, HadGEM3 and observations remains to a rather superficial level. Most of the text presents the content of the figures in a rather descriptive way. This makes the text less attractive and my impression is that section 3 suffers from the lack of interpretation of the results with regards to the paper objectives. Actually, this brings me to the third major concern.

Answer: We agree have amended the introduction to clarify the aim of the paper which is to assess the suitability of the Met Office Unified Model (MetUM) Global Atmosphere (GA) modelling system configurations GA3.0 and GA4.0, two configurations of the HadGEM3 system employed for seamless predictions across climate and weather time scales, for global and continental scales monsoon simulations with realistic features. We place the analysis in the perspective of the model development value chain and the search for avenues for reduced systematic errors in the global monsoon key regional features. The three main objectives of the paper: I. Assess the ability of GA3 and GA4 configurations of the HadGEM3 to capture climatological features of the Global monsoon climate. This includes the influence of physical formulation and horizontal resolution on the HadGEM3 systematic errors. II. Compare GA3 and GA4 performance with

state-of-science CMIP5 outputs; III. Influence of remote versus local circulations on the GA3 and GA4 major systematic errors over the South Asia monsoon domain, and common to many GCMs from the HadGEM family. Since GA3 and GA4 model configuration have been extensively validated in different climate regime, we focus on using the metrics established by the IPCC AR5 for synthesizing and comparing multi-model ensemble performance.

3) My general impression is that the manuscript lacks of focus. Using such an impressive amount of datasets, applied to such a high range of monsoon systems is by all means very welcome. However, this causes the text to attain a hasty character that tries to cover all aspects of a multidimensional analysis. Therefore, it is hard to follow the manuscript. I strongly suggest to the authors to:

Answer: The title of the paper has been modified to clarify the focus of the analysis which reads now as follow : "The global monsoon hydroclimates in HadGEM3 model configurations GA3.0 and GA4.0: Impact of remote versus local circulations errors over South Asia". Furthermore, we have clarified the three main objectives of the paper. The three main objectives of the paper are as follow: I. Assess the ability of GA3 and GA4 configurations of the HadGEM3 to capture climatological features of the Global monsoon climate. This includes the influence of physical formulation and horizontal resolution on the HadGEM3 systematic errors. II. Compare GA3 and GA4 performance with state-of-science CMIP5 outputs; III. Influence of remote versus local circulations on the GA3 and GA4 major systematic errors over the South Asia monsoon domain, and common to many GCMs from the HadGEM family.

i) Be very precise, even by numbering, the objectives in the introduction: The paper content seems to lean towards the model validation aspects, but it also aims "to disentangle the impact of remote versus local atmospheric circulation error". Please provide the objectives and give equal weight to the analysis of each.

Answer: This is noted with appreciation and will be addressed accordingly.

ii) Present all diagnostics and methods in section 2 and link them to the objectives: For instance, several indexes and datasets are first introduced in section 3.

Answer: This is noted with appreciation and will be addressed accordingly.

iii) Present the results in a consistent way, e.g. all diagnostics are applied to all monsoon systems. For instance, it is characteristic that not all the monsoon areas are treated equally. If certain areas need special attention, e.g. Asian or African monsoon, then this could be potentially justified by your results and thus more focused analysis could come in a subsequent section.

Answer: We disagree. While we find the suggestion extremely attractive, it goes beyond the scope of this paper. Please note that HadGEM3 has been already assessed over different individual monsoon regions (e.g West African monsoon, North American monsoon, Asian monsoon). Here we attempt to connect the GA3 and GA4 major systematic errors over the South Asia monsoon domain, also common to some CMIP5 models, with remote versus local circulations in different parts of the global monsoon areas.

My suggestion is either to perform a major revision, or to withdraw the article and resubmit it after adapting the analysis and the text to more precise objectives.

Minor comments Title: At first glance, at least for me, it was not clear if the paper focuses on a specific monsoon region, e.g.: "Hydorclimates of monsoon regions in....".

Answer: We have amended the title which reads as follow: The global monsoon in HadGEM3 model configurations GA3.0 and GA4.0: Impact of remote versus local circulations errors and horizontal resolution.

Answer: We agree with the minor comments bellow and will amend the text accordingly

Abstract: I believe that the abstract would benefit from a more structured presentation. For instance, in line 13 (This work examines..) you could be more precise on the evaluated variables and regions. Then, in the following sentences you could clearly

state the methods that you used (resolutions, nudging, observations...) and finally you could present the most important results and conclusions. Line 16: "Model outputs" instead of "Results"?

Lines 31-33: This phrase seems to be out of context. Maybe it suits better to the beginning of the Abstract.

Line 43-44: "while there is a decreasing trend of lower-troposphere monsoon circulations", this phrase is a bit confusing.

Lines 45: "monsoon coverage" is meant spatial extent of the ITCZ? Could you please clarify. If indeed this refers to the spatial coverage as in line 43, then what is meant by stronger?

Lines 60: " Monsoons are affected by..." The systems/factors mentioned in this sentence have positive and negative feedbacks in precipitation, inland propagation of the rainbands, seasonal rainfall amounts etc. It would be useful to develop these aspects to the extent that the reader acquires a deeper insight into the complexity of the monsoon systems and the challenge faced by the models to properly represent them.

Line 85: Precipitation amounts?

Lines 87-94: A series of phrases refer to interactions, linkages, changes, connections.

Could you please further develop each phrase to be more precise, e.g. what is meant by "volcanic forcing-induced-intensification", which hemisphere? Why is ENSO important for the Australian monsoon?

Line 94: Please remove "climate usually" -> "The West African Monsoon interacts..."

Line 96: This rather long paragraph could benefit from a concluding phrase.

Lines 98-112: I found this part to be of very general nature and less attractive than the following paragraph. I suggest to either omit it, or to make it more relevant to the article's objectives and content.

Lines 131-133 & 135-138. These phrases are complex and hard to understand.

Lines 128- 141: Overall this paragraph lacks of being precise on the paper objectives. Reading the paper up to here, I have a general understanding of the motivation and methods. On the other hand, I am not sure about the paper's contribution to the state of the art and about the methods used to reach them.

Lines 216-220: This part should be moved to the introduction when stating the objectives. Line 253: Why the quotes to "process based statistics"?

Line 272: Please include all datasets used in this study to section 2.4 (e.g. MERRA2 is missing).

Lines 290-292: Please be more informative about IAV and MVI and about their importance for your study.

Line 308: This title is rather too general. Could you please be more precise to the content of the section?

Line 312: AR definition should come before this line. Is it possible to show the monsoon regions in Fig. 4?

Line 327: "The situation improves..." is familiar language. This also stands for most of this paragraph. Please be more specific by quantifying the performed comparisons.

Line 336: replace "3" by "three" and "8" by "eight".

Line 342: Maybe you could skip the paragraph break.

Line 381: As previously, description of the index should come earlier, in the previous sentence.

Figures 4,5 and 12: The geographical coordinates and colorbars use very small font sizes.

Please also note the supplement to this comment:
https://wcd.copernicus.org/preprints/wcd-2020-38/wcd-2020-38-AC1-supplement.pdf
* * *

---

## Author Comment (AC2) · 31 Dec 2020

Dear Reviewer RC2, We are grateful for your careful thoughts and comments and suggestion to implove the paper. Please find below our proposed strategy for revising accordingly the manuscript entitled "The monsoon hydroclimates in HadGEM3 model configurations GA3.0 and GA4.0: Impact of remote versus local circulations errors and horizontal resolution".

[Figure]

Moufouma-Okia et al examined the fidelity of the Met Office Unified Model (MetUM) in simulating the global monsoons climatological features. They have considered the MetUM third and fourth generations Global Atmosphere models and compared the results against multiple observational datasets as well as several atmospheric-only GCMs simulations from the CMIP5 experiments. The improved understanding of the GCM performance is important for studying the variability and projections of global monsoon changes. This is an interesting study on how the remote versus local circulations errors; and the horizontal resolutions of the model influencing the monsoon circulation and related precipitation. However, I would like the authors to address some of my concerns.

Major Comments:

One of the major conclusions given in the abstract and conclusions about the poor simulations of Asian summer monsoon (ASM) by the model which was attributed to excessive precipitation over the southwest equatorial Indian Ocean, rather than to remote tropical atmospheric responses of varying forcing fields, such as SST over the Arabian Sea, aerosols, and growing greenhouse gas emissions. However, this statement was not supported by any kind of analysis. There is no such analysis for the SST over the Arabian Sea, aerosols, and growing greenhouse gas emissions are presented in this manuscript. Also, how the excessive precipitation over the southwest equatorial Indian Ocean contributed to poor simulations of the ASM are not explained.

It is interesting to note that HadGEM3 performance improves significantly over ASM with atmospheric circulations constrained realistically over the tropics, West African and Asian summer monsoon domains. However, it is not clear how exactly that contributed to such performance improvement. Is this due to the improvement of regional low-level circulation or due to the improvement in simulating the vertical wind shears? Are there any differences in the local circulations observed for ASM when atmospheric circulations constrained over the tropics, West African and Asian summer monsoon domains?

Again, you state that the increasing spatial resolution or improving the sub-grid scale parameterizations improves the model simulations (L327-330). This is not always true as in certain cases in your results the simulation doesn't improve even with a higher spatial resolution. The authors need to elaborate more on this.

Answer: We fully agree we the major comments and will amend the text, analysis, conclusion and abstract accordingly.

Minor comments:

Answer: We agree we the minor comments and will amend the text accordingly.

L222: Here it is mentioned that the nudging experiments are carried out through 1982-2008, but you have shown in Table 1 that the integration period for the nudging experiments are 1990-2008. Which is correct? L82-83: "South African monsoon" – is this monsoon over the country South Africa or over the South African region? L260: why do you focus the results only for the period 1984-2005 although your simulation periods are for the period 1982-2008? Fig. 2: Mean and Median of the models - is this the mean for all the CMIP5 models only or the mean for both CMIP5 models and HadGEM3 experiments Fig. 5: Are these results statistically significant? L346-347: "In west Africa, a region. . .." - give a reference. Fig. 8: What does the grey shade indicate? L360-370: Why did you focus only three monsoon regions here? L453: "There is clear benefit in increasing horizontal resolution" – this is not always true. Please explain what exactly improved? Whether the increased horizontal helped improving the simulated precipitation patterns, intensity, etc. Fig. 9 and related results discusses only about the differences in the circulation characteristics, but how do such differences influence the simulated precipitation strength in the models are not discussed. Is the regional monsoon precipitation (e.g. seasonal cycle) in this study calculated only over land or both over land/ocean? Fig. 12: Domain Def: Psummer – Pwinter > 2. Is this 2.5 instead of 2 as per L78?

Please also note the supplement to this comment:
https://wcd.copernicus.org/preprints/wcd-2020-38/wcd-2020-38-AC2-supplement.pdf
* * *

---

## Editor Comment (EC1) · Heini Wernli (Editor) · 4 Jan 2021

Dear Wilfran Moufouma and co-authors

I had a look at your final author comments. They are very brief and seem to be written in a rather hastily way. If you plan to submit a revised version of your paper, then It will be very important that the accompanying reply document is much more detailed and carefully written.

Reviewer 1 mentions language issues; they unfortunately also appear in the final au-

thor comments. Several sentences are difficult to understand, e.g., "We place the analysis in the perspective of the model development value chain and the search for avenues for reduced systematic errors in the global monsoon key regional features." or "Impact of remote versus local circulations errors over South Asia" - is it "impact of remote and local circulation errors on the representation of the South Asian monsoon"? or is it "Impact of remote and local circulations on errors in the South Asian monsoon"? For my assessment of the revised version, language and clarity of the text will be essential.

It is also confusing that you write twice on the same page that you plan to change the title of the paper, but the new titles you mention are not the same.

With best regards, Heini Wernli (co-editor)